# Sensitivity Analysis of Cardiac Alternans and Tachyarrhythmia to Ion Channel Conductance Using Population Modeling

**DOI:** 10.3390/bioengineering9110628

**Published:** 2022-11-01

**Authors:** Da Un Jeong, Aroli Marcellinus, Ki Moo Lim

**Affiliations:** 1Department of IT Convergence Engineering, Kumoh National Institute of Technology, Gumi 39253, Korea; 2Department of Medical IT Convergence Engineering, Kumoh National Institute of Technology, Gumi 39253, Korea

**Keywords:** cardiac alternans, tachyarrhythmia, population modeling, sensitivity analysis, population modeling

## Abstract

Action potential duration (APD) alternans, an alternating phenomenon between action potentials in cardiomyocytes, causes heart arrhythmia when the heart rate is high. However, some of the APD alternans observed in clinical trials occurs under slow heart rate conditions of 100 to 120 bpm, increasing the likelihood of heart arrhythmias such as atrial fibrillation. Advanced studies have identified the occurrence of this type of APD alternans in terms of electrophysiological ion channel currents in cells. However, they only identified physiological phenomena, such as action potential due to random changes in a particular ion channel’s conductivity through ion models specializing in specific ion channel currents. In this study, we performed parameter sensitivity analysis via population modeling using a validated human ventricular physiology model to check the sensitivity of APD alternans to ion channel conductances. Through population modeling, we expressed the changes in alternans onset cycle length (AOCL) and mean APD in AOCL (AO meanAPD) according to the variations in ion channel conductance. Finally, we identified the ion channel that maximally affected the occurrence of APD alternans. AOCL and AO meanAPD were sensitive to changes in the plateau Ca^2+^ current. Accordingly, it was expected that APD alternans would be vulnerable to changes in intracellular calcium concentration.

## 1. Introduction

Action potential duration (APD) alternans refers to beat-to-beat alternations of an action potential. It can evolve into atrial fibrillation and ventricular fibrillation as precondition for predicting the likelihood of cardiac arrhythmias [1,2]. The APD alternans mechanism was identified using graphical methods generated by simulations [3,4]. These methods use APD restitution (APDr) curves based on the shortening of APD in myocardial cells with rapid heart rate and show APD’s dependence on the prior diastolic interval. Advanced studies have validated that as heart rate becomes faster, the likelihood of generation of APD alternans increases through APDr curves [3,4]. However, some patients with atrial fibrillation in clinical trials also developed APD alternans, even in slow-beat conditions with near-normal heart rates [5,6].

In patients with typical atrial flutter, Narayan et al. measured monophasic action potential. They confirmed that APD rate maladaptation at the isthmus could cause APD alternans or electrical conduction blockages in tissue, thus transforming into atrial fibrillation [7]. The dynamic substrates of atrial fibrillation can be identified via APD alternans. Based on the clinical trials, APD alternans did not occur in control groups without atrial fibrillation. Still, they happened at a lower rate and higher amplitudes in persistent atrial fibrillation than in paroxysmal atrial fibrillation [6]. The tests confirmed that ectopic beats induced atrial fibrillation in patients with APD alternans at a slow rate. In patients with APD alternans at a fast pace, irregular movements of action potentials affected the generation of atrial fibrillation. Accordingly, Narayan et al. suggested the need to identify the mechanisms by which APD alternans were generated at heart rates close to the resting state [6]. Chang et al. investigated the causes and mechanisms of APD alternans generation at slow heart rate through a simulation using an ion model for human atrial cells and a three-dimensional cubic tissue model [5,8]. They found that a load of intracellular calcium concentrations increased when the inactivation rate of ryanodine receptors (RyR) present in the sarcoplasmic reticulum (SR) decreased [5]. Furthermore, through atrial fibrillation simulations using a three-dimensional human atrial model, Chang and Trayanova demonstrated that calcium-driven APD alternans could be transformed into arrhythmias such as atrial fibrillation [1].

Some studies have identified APD alternans in terms of electrophysiological ion channel currents in cells. Fox et al. performed the simulation study using the newly implemented ion model of myocardial cells, based on direct experimental results and previous validated ionic current models. They confirmed that when the amplitude of calcium currents or calcium-driven inactivation rates decreased and the amplitude of potassium currents (IKi, inward rectifier current; IKr, rapid-delayed rectifier current; IKs, slow-delayed rectifier current) increased, the development of APD alternans could be inhibited [9]. Sobie confirmed that sensitivity to APD alternated according to changes in electrical conductivity in ion channels and predicted physiological phenomena such as action potential or intracellular calcium concentration through partial least squares (PLS) regression. Sobie predicted the peak and resting values of membrane potential and APD by randomly changing each ion channel conductance using a PLS regression model [10]. He finally confirmed that changes in action potentials were most affected by calcium channel currents and three potassium channel currents, which corresponded to the findings of Fox et al. [9] and Kurata et al. [11]. 

The studies mentioned above have identified physiological phenomena such as action potential owing to random changes in a particular ion channel’s conductivity through ion models. For their purpose, they generally used models specialized in specific ion channel currents. However, they did not confirm which combination of changes in each ion channel induced or affected APD alternans. This study performed parameter sensitivity analysis via population modeling using a validated human ventricular electrophysiology model to check the sensitivity of ion channels to APD alternans. Through population modeling, we expressed the changes in alternans onset cycle length (AOCL), which is the cycle length of first-time APD alternans, and the average APD of 10 beats at the AOCL (AO mean APD) according to the variations in ion channel conductance. Finally, we identified the ion channel that mainly affects the occurrence of APD alternans.

## 2. Materials and Methods

### 2.1. In Silico Electrophysiological Model

In this study, we used the validated human ventricular epicardial cell model proposed by Ten Tusscher et al. in 2004 to confirm the sensitivity to APD alternans [12]. The membrane potential (Vm) in myocardial cells was implemented using the following electrical conduction equation:(1)$$\frac{dV_{m}}{dt}=-\frac{I_{ion}+I_{stim}}{C_{m}}$$
(2)$$I_{ion}=I_{Na}+I_{K1}+I_{to}+I_{Kr}+I_{Ks}+I_{CaL}+I_{NaCa}+I_{NaK}+I_{pCa}+I_{pK}+I_{bCa}+I_{bNa\;\;}$$
where *C_m_* refers to the capacitance for the unit surface area of the cell membrane and is set at 2.0 uF/cm^2^, *I_ion_* represents the sum of the ion currents through the cell membrane, and *I_stim_* represents the current caused by external stimulation. *I_ion_* is composed of *I_Na_*, fast Na^+^ current; *I_K_*_1_, inward K^+^ current; *I_to_*, transient outward K^+^ current; *I_Kr_*, rapid delayed rectifier K^+^ current; *I_Ks_*, slow delayed rectifier K^+^ current; *I_CaL_*, L-type Ca^2+^ current; *I_NaCa_*, Na^+^-Ca^2+^ exchange current; *I_NaK_*, Na^+^-K^+^ exchange current; *I_pCa_*, plateau Ca^2+^ current; *I_pK_*, plateau K^+^ current; *I_bCa_*, background Ca^2+^ leakage current; and *I_bNa_*, background Na^+^ current. All ion channel currents were represented by the electrical conductance (G), membrane potential, and equilibrium potential for each ion channel based on the Hodgkin–Huxley equation (Appendix A). To observe the sensitivity of cardiac alternans to ion channel conductance, we reduced the electrical conductance of each ion channel by 50% and then increased it by 50% from the normal condition of default conductance constants; in Table 1, the 100% scale denotes the control condition where no changes have occurred and the 50% and 150% scales denote the decreased case to 50% and increased case to 150%, respectively. The scaling values of the conductances were set based on physiological and pathological conditions; for example, the g_Ks_ values scales were set to 10%, denoting a reduction of g_Ks_ by 90% from the normal situation to mimic the intermediately revealed state of the S140G mutation condition, which is well-known to induce atrial fibrillation [13]. Another gain-of-function mutation, the L450F mutation leading to long QT syndrome and Brugada syndrome, increased the g_to_ value of the ventricular myocyte by 78%, which was experimentally derived by Giudicessi et al. [14]. By combining 10 ion channels with 3 ion channel conductance conditions, we generated 59,049 scenarios of ion channel conductance variations.

### 2.2. Simulation Protocols

The occurrence of APD alternans under various ion channel conductance scenarios was simulated using clinical pacing protocols, as suggested by Narayan et al. to identify the generation of APD alternans in atrial fibrillation patients [6]. Cardiomyocytes were initialized with a steady-state condition at a 750 ms cycle length. We decreased the cycle length in 50 ms decrements, generating 74 pacings for each cycle length. When the cycle length reached 350 ms, we set the decrement to 10ms until the cycle length was 180 ms and generated 74 pacings.

We worked out the electrophysiological simulations using the human ventricular myocyte model according to the variations of ionic channel conductance. Then, for each scenario of conductance variations, we identified and analyzed the occurrence of APD alternans through changes in membrane potential over time. The occurrence of APD alternans was confirmed by calculating APD alternans normalization magnitude (ANM), as shown in the following formula:(3)$$\mathrm{ANM}=\frac{AM}{meanAPD}$$
(4)$$\mathrm{AM}=\frac{\sum_{i=1}^{10}∆\left(m_{i+1}-m_{i}\right)}{10}$$
where AM is obtained by averaging the differences in the amplitude of the AP in the last 10 beats (from 11 beats) under each cycle length condition and m_i_ represents the amplitude of the action potential in the i-th beat. APD was considered based on APD90, which measures the period from depolarization of cardiomyocytes to 90% repolarization after the maximum upstroke. The average APD90 (meanAPD) was calculated from the last 10 beats, and the ratio of AM to meanAPD was defined as ANM. When ANM was greater than 0.05, APD alternans occurred. Accordingly, we defined the longest cycle length (>0.05 of ANM) as the AOCL, in which APD alternans first appeared. Furthermore, the AO meanAPD was determined as meanAPD in AOCL [6].

### 2.3. Population Modeling

The population map proposed by Gemmell et al. was used to efficiently express the changes in the AOCL and AO APD according to ion channel conductance variations [15]. 

The population map was expressed by accumulation of elements of the smallest scale internally from the largest scale outside (Figure 1). Ion channel conductances, in the label of each scale, were assigned in random order. Then, using optimization algorithms, we placed ion channel conductances in order on the outermost largest scale, which had the greatest impact on AOCL and AO APD [4,15]. To optimize the population map, we calculated the sum of the differences between the scalar value of a specific node (scenario) and scalar values of adjacent nodes in the horizontal and vertical directions in a permutation. Depending on the number of cases where 10 ion channel conductances could be arranged, the permutation was 10!/2 in total, considering that the x-axis and y-axis arrangements were symmetrically identical. We finally generated the optimized population map by obtaining the configuration where the sum of the differences in the 59,049 scenarios was the lowest [16,17].
(5)$$\mathrm{Optimization}\;\mathrm{Map}=\mathrm{minimum}\left\{\overset{all\;scenarios}{\underset{i=1}{\sum }}\left(\left|s_{h1}-s_{i}\right|+\left|s_{v1}-s_{i}\right|+\left|s_{h2}-s_{i}\right|+\left|s_{v2}-s_{i}\right|\right)\right\}$$
where i denotes a scenario under a specific combination condition of 10 ion channel conductance variations and *s_i_* refers to a scalar value of AOCL or AO APD in scenario i. s_h1_, s_h2_, s_v1_, and s_v2_ represent the scalar values in the horizontally and vertically adjacent nodes in s_i_, respectively.

## 3. Results

### 3.1. The Most Influential Ion Channels to AOCL and AO MeanAPD

Figure 2 shows the changes in AOCL and AO meanAPD according to the variation of the 10 ion channel conductances. The population map in Figure 2 was adjusted by placement of the ion channel with the maximum influence on each dependent variable on the outermost axis. AOCL responded most sensitively to the change in I_pCa_ due to the g_pCa_ variations and the change in I_CaL_ due to the g_caL_ variations. The population map of AO meanAPD according to the variations in ion channel conductance was different in the positions of g_CaL_, g_Kr_, and g_k1_ compared to that of AOCL. Accordingly, AO meanAPD was the most sensitive to changes in g_K1_ as well as changes in I_pCa_ due to changes in g_pCa_.

### 3.2. Ion Channel Variation Scenarios at the Maximal and Minimal AOCL

AOCL was maximal at 550 ms in 81 cases of the 59,049 ion channel conductance variation scenarios. In those 81 scenarios, g_CaL_ and g_bCa_ increased by 150% compared to the original electrical conductance, and g_pCa_ as well as three potassium channel current conductances of g_Ks_, g_Kr_, and g_K1_ decreased by 50% compared to the original (Figure 2a). The AO meanAPD for those 81 scenarios was 474.3 (461.1–490.4) ms. Figure 3a shows the APD restitution curves in the scenario where AO meanAPD was maximal at 490.4 ms and minimal at 461.1 ms among the 81 scenarios with the longest AOCL. AO meanAPD was maximal in the variation scenario where g_caL_ and g_bca_ increased by 50% and other conductances decreased by 50%. AO meanAPD was minimal when g_bNa_, g_caL_, g_bCa_, and g_pK_ increased to 150% and when g_Ks_, g_Kr_, g_K1_, g_Na_, g_to_, and g_pCa_ decreased by 50%. In both scenarios, the APD became longer as the cycle length decreased. APD alternans was not observed until the cycle length was 600 ms, and the ANMs at a cycle length of 600 ms in both scenarios were 0.00056 and 0.0000, which are less than 0.05. In both scenarios, APD alternans first occurred when the cycle length of the cardiomyocyte was shortened to 550 ms and was observed in the form of repeating odd-numbered beats with large amplitudes and even-numbered beats with small amplitudes. At this time, ANM was 0.1853 when AO meanAPD was maximum and 0.3138 when it was minimum (Figure 3b,c).

There were 2817 ion channel conductance variations scenarios with a minimal AOCL of 180 ms, and the average AO meanAPD was 106.3 (60.05–155.7) ms. Figure 3d shows the APD restitution curves when the AO meanAPD was at the maximum and the minimum among these minimum AOCL scenarios. Among the smallest AOCL conditions, when g_Ks_, g_Na_, and g_pCa_ increased by 150% and g_CaL_, g_bCa_, and g_pK_ decreased by 50%, the AO meanAPD was the greatest. The AO meanAPD was the shortest when g_pCa_ and g_pK_ decreased by 50% and other conductances increased by 150% except for g_K1_. In the former scenario, even when the cycle length was 190 ms, beats with large and small amplitudes were repeatedly observed, but the algorithm used did not determine these action potentials as APD alternans (ANM = 0.0455). The difference in repetitive beats of the action potentials increased more when the cycle length was 180 ms than when it was 190 ms and was eventually judged to be APD alternans (ANM = 0.0553, Figure 3e). In the latter scenario, when the cycle length was more than 400 ms, the APD was longer than in the scenario with the maximum AO meanAPD under the minimal AOCL condition. However, when the cycle length was shortened to 400 ms or less, the APD sharply decreased; thus, large and small beats were repeatedly generated when the cycle length was 180 ms. At that time, the ANM was 0.3129 (Figure 3f).

### 3.3. Ion Channel Variation Scenarios at the Minimal AO meanAPD

As for the AO meanAPD, the longer the AOCL was, the longer the AO meanAPD was according to the changes in ion channel conductance. Accordingly, there was a high statistical correlation between the two variables (correlation coefficient = 0.958, *p*-value < 0.01). Figure 3g,h show the APD restitution curves and action potential when the AO meanAPD was at the minimum of 56.7 ms among all ion channel conductance variation combinations. In that scenario, g_Ks_, g_Na_, and g_pCa_ decreased by 50% and other ion channel conductances (g_Kr_, g_K1_, g_bNa_, g_CaL_, g_bCa_, g_to_, and g_pK_) increased by 150%. According to the reduction in cycle length, the APD changes in that scenario were almost similar to those of the minimum AD meanAPD scenario when the AOCL was minimum. The APD decreased rapidly when the cycle length was 350 ms or less (Figure 3g). When the cycle length was 250 ms, the APD gradually decreased from 61.0 ms to 60.9 in the last 10 beats and the ANM was 0.000164. When the cycle length reached 240 ms, APD alternans was first observed in the last 10 beats, in which the beat with an APD of 66.3 ms and the beat with an APD of 48.8 ms were alternately generated (ANM = 0.3083, Figure 3h).

### 3.4. Non Alternans Scenarios

When APD alternans did not occur even though the alternans generation protocol was applied, both AOCL and AO meanAPD had a value of 0. There were 8437 scenarios in which APD alternans did not occur (Appendix A). Scenarios where APD alternans did not occur are displayed in gray on the population map. These cases are distributed in the right diagonal direction on the AOCL population map and concentrated on the right side of the AO meanAPD population map as a whole.

## 4. Discussion

In this study, we identified the ion channel that was most affected by AOCL and AO meanAPD when APD alternans occurred through population modeling. The main findings of this study are as follows:AOCL and AO meanAPD were sensitive to changes in the plateau Ca^2+^ current (g_pCa_). Accordingly, it was expected that APD alternans would be vulnerable to changes in intracellular calcium concentration.When APD alternans occurred, AOCL and AO meanAPD were proportional correlations (correlation 0.958, *p*-value < 0.01), similar to the cardiomyocytes without APD alternans; it is well-known that APD decreases as the cycle length shortens in normal cardiomyocytes.

The existing hypotheses describing the mechanism of cardiac fibrillation are related to the recovery of action potentials. The APD was plotted according to changes in the cycle length and diastolic interval. It then determined the occurrence of APD alternans or electrical instability through the slope (more than 1) of restitution curves [3,4,18]. However, in some patients with atrial fibrillation, even though the slope of the APD restitution curve was less than 1, APD alternans occurred in the cycle length close to the resting state [6]. In this study, among 59,049 combinations of ion channel conductance variations, the cardiomyocytes of the scenarios in which APD alternans occurred at a slow heart rate of 100–120 bpm (cycle length 600–500 ms) exhibited an unstable state in which the APD became longer as the cycle length was shortened. Accordingly, it was expected that APD alternans would occur under conditions of a slow heart rate. The graph also shows the recovery of the action potential from a cycle length of 2000 ms to 180 ms (Appendix A). When APD alternans occurred at slow heart rate conditions, the slope of the APD restitution curve was more than −1 (it decreased more steeply). In those cases, the APD alternans generated in the tissue was transferred to the collapse of the reentrant waves, leading to more unstable fibrillation [19,20].

APD alternans occurring under slow heart rate conditions differed from that arising under fast heart rate conditions. APD alternans occurring at fast heart rates was strongly related to sodium channel currents [4], whereas at slow heart rates, it was closely associated with the calcium channel current [1,5]. Using a clinical pacing protocol, we observed that APD alternans occurred at cycle lengths close to the resting heart rate in several scenarios of ion channel conductance variations. It was confirmed that the occurrence of APD alternans was related to calcium channel currents; the most influential channel conductance was g_pCa_. In particular, when APD alternans occurred at a cycle length close to the resting state, there were changes mainly in the conductances of three calcium current channels: g_caL_, g_pCa_, and g_bCa_ (Figure 2a). This is because the amount of calcium entering the cytoplasm from the outside of the cell increased due to the increased I_CaL_ (g_caL_) and the increased I_bCa_. However, the amount of calcium excreted from the cytoplasm out of the cell decreased due to the decrease in I_pCa_ (g_pCa_). Accordingly, it was expected that APD alternans would occur at a slow heart rate because of the prolonged APD and the intracellular calcium concentration [6,21,22]. These results correspond to the experimental results of Chang et al.; they found that cellular electrical instability and APD alternans at a slow heart rate could occur through changes in the calcium concentration released and updated in the sarcoplasmic reticulum, inside cells, caused by decreased inactivation of ryanodine receptors [5].

The average APD in the last 10 beats at the AOCL was sensitive to the change in the I_K1_ current (g_K1_). In general, the I_K1_ current played an important role in stabilizing the plateau at the action potential, terminating repolarization, and maintaining the exciting membrane potential to rest [23]. In cells in the resting state, decreasing I_K1_ current could lead to longer APD, which could cause long QT intervals in the ventricular tissue [24]. These dynamic mechanisms worked similarly even when APD alternans occurred, suggesting that APD at the AOCL lengthened as g_K1_ decreased. In this study, the changes in AOCL and AO meanAPD were greater with changes in calcium current conductance than with changes in potassium current conductance. This was expected because the dynamics of the inward rectifier current did not directly affect the APD, which changes according to the cycle length of cardiomyocytes, but could indirectly affect APD by inducing changes in the ion currents due to the change in the calcium and potassium concentration inside the cell [25]. Clinical experiments performed by Ravens and Cerbai have shown that delayed after-depolarization caused by increased intracellular calcium concentrations can increase the likelihood of ventricular tachycardia in non-ischemic heart failure patients and a decrease in I_K1_ can increase this possibility [24].

We determined the APD alternans conditions when ANM was over 0.05 by following the advanced studies of Chang et al. [5] and Narayan et al. [6]. They decided on the ANM threshold that determines the alternans condition through the experimental model and simulations; Chang et al. especially validated it by observing the significant alternans when ANM was over 0.05, even if some alternans had occurred at less than 0.05 ANM [5]. Accordingly, our study also observed several conditions that appeared to be alternans at the ANM condition of less than 0.05, as shown in Figure 3e. Those were not significant alternans decided by the ANM threshold but evolved into prominent alternans.

This study expressed the sensitivity of the 10 ion channel electrical conductance to APD alternans through a population map. The population modeling used the dimensional stacking method on a map of only two dimensions to represent multi-dimensional data efficiently by reducing visual complexity [16,17]. It was a challenge to visually distinguish the sensitivity of the generated population map to specific response variables before rearranging the dimensions of each axis. Appendix A shows the population map before rearrangement of the dimensions. Therefore, to determine the ion channel conductance that affected AOCL and AO meanAPD-quantified APD alternans, dimensions were rearranged to the optimized coordinates by minimizing the sum of differences between the value of a specific coordinate and the values of adjacent coordinates [15,16,26].

Inter-individual variability crucially affects the progression of diseases and treatment [27], hence several studies have considerably advanced the population model regarding the effects of inter-individual variability on physiological phenomena. Britton et al. developed a coupled model of experimental measurements and mathematical modeling to calibrate the myocyte model according to inter-individual physiological variability [28]. Song et al. investigated how I_NaK_ plays a significant role in the maximal slope of APD restitution curves through sensitivity analysis using the atrial myocyte population model and the interpretable machine learning model [29]. Llopis–Lorente suggested that the population model could improve the assessment accuracy of drug-induced torsadogenic risk by choosing different biomarkers depending on individuality [30].

There was a limitation to this study. We only considered the APD alternans occurring in the single-cell level. APD alternans can occur at the tissue level, and anisotropy plays an important role in the development of APD alternans by affecting the occurrence of APD alternans [31,32,33]. In addition, the ventricular tissue has heterogeneous characteristics: the myocyte of each tissue, such as the endocardium, mid-myocardium, and epicardium; and different conductances of ion channels [29,33]. However, it was impossible to consider the heterogeneity of ventricular tissue through the single-cell model. Therefore, it is necessary to check the sensitivity of APD alternans in consideration of the characteristics in the ventricular tissue model.

## 5. Conclusions

The population maps of AOCL and AO meanAPD as well as the study results can predict the occurrence of APD alternans under various scenarios according to the changes in the ion channel conductance. This can identify the occurrence of APD alternans near resting heart rate, which is observed clinically, as well as inducing APD alternans, which is commonly known to occur under fast heart rate conditions. Therefore, it can be used to infer the ionic mechanisms that would be transferred to atrial fibrillation. In addition, the results of this study may help predict the occurrence of APD alternans under various drug conditions.

## Figures and Tables

**Figure 1 bioengineering-09-00628-f001:**
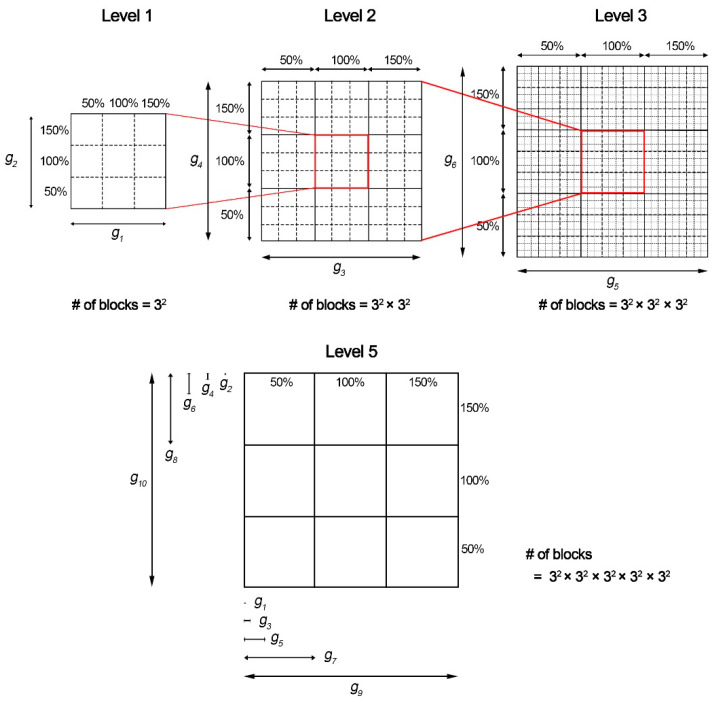
Population modeling process: g_#_, the conductances of 10 ion channels.

**Figure 2 bioengineering-09-00628-f002:**
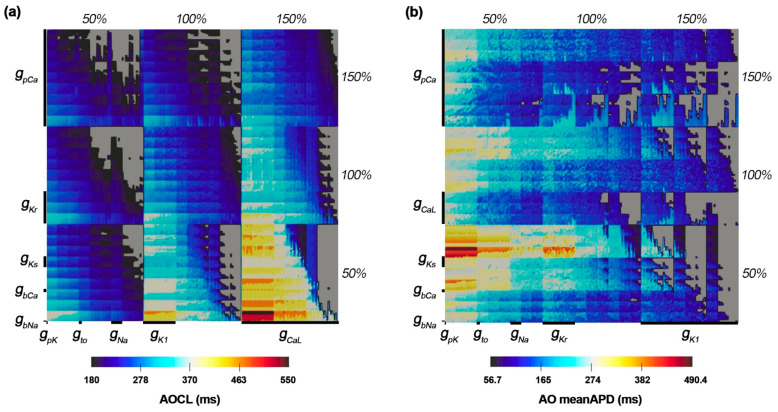
(**a**) Population maps of AOCL and meanAPD at alternans onset. AOCL is alternans onset cycle length; (**b**) AO meanAPD is an average of action potential duration in AOCL. Gray parts represent non APD alternans scenarios.

**Figure 3 bioengineering-09-00628-f003:**
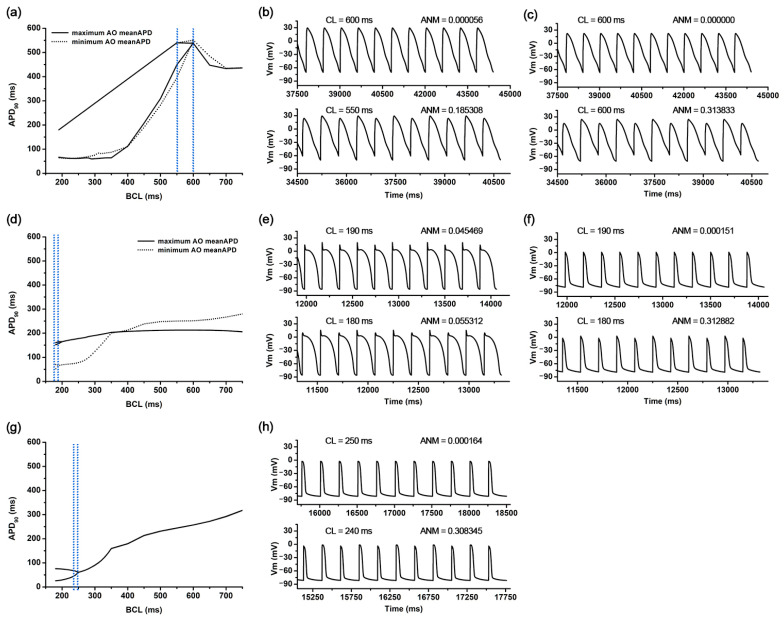
APDr curves and AP shapes under the representative scenarios: APDr curves in the scenarios of the longest meanAPD and shortest meanAPD in the longest AOCL (**a**), AP shapes of the former scenario (**b**), and AP shapes of the latter scenario (**c**); APDr curves in scenarios of the longest meanAPD and shortest meanAPD in the shortest AOCL (**d**), AP shapes of the former scenario (**e**), and AP shapes of the latter scenario (**f**); APDr curves in a scenario of the shortest AO meanAPD alternans (**g**) and corresponding AP shapes (**h**); AP shapes before AOCL; and AOCL. APDr is action potential restitution curve; AOCL, alternans onset cycle length; AP, action potential; AO meanAPD, an average of action potential duration in AOCL; CL, cycle length; and ANM, APD alternans normalization magnitude.

**Table 1 bioengineering-09-00628-t001:** Initial conductances of 10 ion channels based on the human ventricular myocyte model of Ten Tusscher et al. [12].

Abbreviation	Description	Conductance (nS/pF)
G_Ks_	Conductance of slow delayed rectifier K^+^ current	0.245
G_Kr_	Conductance of rapid delayed rectifier K^+^ current	0.096
G_K1_	Conductance of maximal inward K^+^ current	5.405
G_Na_	Conductance of maximal Na^+^ current	14.838
G_bNa_	Conductance of maximal background Na^+^ current	0.00029
G_CaL_	Conductance of maximal L-type Ca^2+^ current	0.0000398
G_bCa_	Conductance of maximal background Ca^2+^ current	0.000592
G_to_	Conductance of transient output K^+^ current	0.294
G_pCa_	Conductance of maximal plateau Ca^2+^ current	0.025
G_pK_	Conductance of maximal plateau K^+^ current	0.0146

## Data Availability

Not applicable.

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
