# Peer review of "Sensitivity Analysis of Cardiac Alternans and Tachyarrhythmia to Ion Channel Conductance Using Population Modeling"

_bioengineering, 2022, doi:10.3390/bioengineering9110628_

Round 1
Reviewer 1 Report
Jeong et al. conduct a study exploring the relationship between ion channel conductances and action potential duration (APD) alternans occurrence. They utilize metrics such as APD alternans onset cycle length(AOCL) and mean APD at AOCL(AO meanAPD) to quantify the relationship and to identify the conductance parameter with the highest impact on alternans occurrence.
The authors find that AOCL and AO meanAPD are most sensitive to the GpCa(Ca2+ plateau current) parameter via a population map optimization technique, and that AOCL and AO meanAPD have a high positive correlation with each other. These findings signify that intracellular calcium levels can significantly affect APD alternans onset, and that APD alternans can potentially occur at resting heart rate as well as slow and fast heart rates.
Overall, the manuscript was well-written and the premise of this study to investigate the onset of alternans at different cycle lengths was interesting. The use of the population map optimization provided an intuitive method of interpreting the parameter sensitivity analysis. However, the range of conductance parameter variations is missing a physiological justification and should be supported accordingly in a similar manner to Gemmell et al.[13], one of the cited works.
Major critiques:
● A physiological rationale is not given for the range of conductances. Are they within the bounds of experimentally obtained values in cardiomyocytes? Is the full range of possible variations explored?
● The conductances are decreased by 50%(x0.5), but increased by 150%(x2.5), which are different intervals. Do the authors mean to say increase to 150%(therefore an increase by 50%), or are the different intervals intentional? If intentional, why was this difference created?
● The Level 3 grid in Figure 1 is different from Gemmell et al.’s work. Compared to their figures, the Level 3 grid shows incorrect scaling and location of its Level 2 representation. The representation scale is actually at the size of Level 1, and the stacking location is not in the middle of the Level 3 grid.
● Authors claim that alternans occur when ANM>0.05 in Materials and Methods(Lines 132~133). However, there is no rationale given for this value. Also, they acknowledge that they observed alternan-like behavior(Figure 3e) at ANM=0.0455(Line 194), which is close to 0.05 but is still contradictory to their claim.
● The significance of Finding #2 in the Discussion(Line 232) is unclear, and could be explained in more detail in the Discussion.
Minor edits:
● Line 230: Authors incorrectly indicate plateau Ca2+ current conductance as GCaL, when it should be GpCa.
● Inconsistent definition vs. usage of AO meanAPD. In Lines 81~82, the definition of AO meanAPD is given as the mean of “change” in APD, while its usage in Lines 134~135 show that it is being used as the mean APD itself.
● Inconsistent naming convention of variables. For instance, the word “mean” is used normally in AO meanAPD, while subscripted in APDmean. Also, APD90 and APD90 are simultaneously used, etc.
● Line 69: The conductance randomization process was not done by the PLS algorithm itself. It was actually conducted by randomly choosing from a log-normal distribution. Refer to Sobie’s paper for more information.
● Line 115: Myocardiocytes -> Cardiomyocytes
● Line 116~117: The meaning of this sentence is ambiguous, and should be re-worded for clarification. Currently, it can be interpreted as either:
A) The authors conducted 74 pacings in total as they decreased the cycle length.
B) The authors conducted 74 pacings for each cycle length value.
● Line 118~119: Same comment as 116~117.
● Line 116~119: Several grammar mistakes caused by referring to “cycle length” as a living object. For example, a cycle length cannot “give pacings”, nor can it by itself “reduce cycle length”.
● Line 127: “amplitude of APD” -> “amplitude of AP”
● Line 127~128: The phrase “last ten pairs of beats” suggests there are 20 beats, which is inconsistent with the provided equation which suggests 10 beat differences(from 11 beats) are used. Authors should re-word this phrase for clarity and consistency.
● Line 133: Unfinished parentheses
● Line 151: senarios -> scenarios
Author Response
We appreciate the reviewer’s and editor’s comments, and we have revised the manuscript accordingly. Please see the attachment.

Reviewer 2 Report
The manuscript addresses a key topic of the current cardiac research. The work performed is worth publication and of interest to a general audience. However, I think the authors partially missed two critical aspects that I ask to complete before accepting the manuscript for publication.
1. It is mandatory to link this work with recent contributions from outstanding groups (e.g., Rodriguez & Bueno-Orovio Oxford as a representative example) that proposed advanced techniques based on deep learning schemes.
2. Considering the mentioned limitations, I think it is much more important to introduce the concept of dispersion of repolarization other than mechanoelectric feedback. In fact, there is a vast literature not mentioned here that has demonstrated the critical role of cell-cell nonlinear coupling, material heterogeneity and anisotropy concerning the onset and development of APD alternans (e.g., https://www.frontiersin.org/articles/10.3389/fphys.2013.00071/full, https://www.sciencedirect.com/science/article/abs/pii/S0045782515003679, https://journals.aps.org/pre/abstract/10.1103/PhysRevE.87.042717, https://aip.scitation.org/doi/abs/10.1063/5.0050897)
Author Response

(The authors gave the same response as above.)
